# Characterization of QuantiFERON-TB-Plus Results in Patients with Tuberculosis Infection and Multiple Sclerosis

**DOI:** 10.3390/neurolint17080119

**Published:** 2025-08-02

**Authors:** Elisa Petruccioli, Luca Prosperini, Serena Ruggieri, Valentina Vanini, Andrea Salmi, Gilda Cuzzi, Simonetta Galgani, Shalom Haggiag, Carla Tortorella, Gabriella Parisi, Alfio D’Agostino, Gina Gualano, Fabrizio Palmieri, Claudio Gasperini, Delia Goletti

**Affiliations:** 1Translational Research Unit, National Institute for Infectious Diseases Lazzaro Spallanzani-IRCCS, 00149 Rome, Italy; elisa.petruccioli@inmi.it (E.P.); valentina.vanini@inmi.it (V.V.); andrea.salmi@inmi.it (A.S.); gilda.cuzzi@inmi.it (G.C.); 2Department of Neurosciences, San Camillo Forlanini Hospital, 00152 Rome, Italy; luca.prosperini@gmail.com (L.P.); serena.ruggieri@gmail.com (S.R.); galgasi@tiscali.it (S.G.); neuroshalom@hotmail.com (S.H.); carla.tortorella@gmail.com (C.T.); cgasperini@scamilloforlanini.rm.it (C.G.); 3Unità Operativa Semplice (UOS) Professioni Sanitarie Tecniche, National Institute for Infectious Diseases Lazzaro Spallanzani-IRCCS, 00149 Rome, Italy; 4Department of Clinical Microbiology and Virology, San Camillo Forlanini Hospital, 00152 Rome, Italy; gabriella.parisi@scamilloforlanini.rm.it (G.P.); adagostino2@scamilloforlanini.rm.it (A.D.); 5Respiratory Infectious Diseases Unit, National Institute for Infectious Diseases Lazzaro Spallanzani-IRCCS, 00149 Rome, Italy; gina.gualano@inmi.it (G.G.); fabrizio.palmieri@inmi.it (F.P.)

**Keywords:** QFT-Plus, IFN-γ, multiple sclerosis, MS, tuberculosis

## Abstract

Background: Disease-modifying drugs (DMDs) for multiple sclerosis (MS) slightly increase the risk of tuberculosis (TB) disease. The QuantiFERON-TB-Plus (QFT-Plus) test is approved for TB infection (TBI) screening. Currently, there are no data available regarding the characterization of QFT-Plus response in patients with MS. Objectives: This study aimed to compare the magnitude of QFT-Plus responses between patients with MS and TBI (MS-TBI) and TBI subjects without MS (NON-MS-TBI). Additionally, discordant responses to TB1/TB2 stimulation were documented. Results were evaluated considering demographic and clinical data, particularly the impact of DMDs and the type of TB exposure. Methods: Patients with MS (N = 810) were screened for TBI (2018–2023). Thirty (3.7%) had an MS-TBI diagnosis, and 20 were recruited for the study. As a control group, we enrolled 106 NON-MS-TBI. Results: MS-TBI showed significantly lower IFN-γ production in response to TB1 (*p* = 0.01) and TB2 stimulation (*p* = 0.02) compared to NON-MS-TBI. The 30% of TB2 results of MS-TBI fell into the QFT-Plus grey zone (0.2–0.7 IU/mL). Only 7% of NON-MS-TBI showed this profile (*p* = 0.002). Conclusions: MS-TBI had a lower QFT-Plus response and more borderline results compared to NON-MS-TBI. Future studies should clarify the significance of the borderline results in this vulnerable population to improve QFT-Plus accuracy regarding sensitivity, specificity, and TB prediction.

## 1. Introduction

*M. tuberculosis* (Mtb) was responsible for 10.8 million new cases of tuberculosis (TB) disease and 1.3 million deaths in 2023 [1]. It has been estimated that 1.7 billion persons have a tuberculosis infection (TBI) defined by an immune response to Mtb without signs or symptoms of disease, representing almost a quarter of the world population [2]. About 20–25% of people exposed to Mtb become infected; among them, 5–10% could develop TB disease during their lifetime [3,4,5,6]. Ninety percent of people with an immune response to Mtb control its replication. However, Mtb may persist for years in the host in a quiescent status, continuously stimulating the immune system [7,8,9,10,11,12]. Individuals with an impaired immune response, such as people living with HIV [13,14] or treated with specific drugs acting on the immune system, such as persons with immune-mediated inflammatory disease [15], have a higher risk of developing TB disease [16]. Autoimmune diseases have been increasingly recognized as comorbidities in individuals with TBI. The interplay between chronic immune activation and immune dysregulation may predispose patients with autoimmune diseases to progression to TB disease, especially when treated with immunosuppressive therapies, as already demonstrated for rheumatological disease [17,18,19,20,21,22]. This is particularly relevant in the context of MS, where disease-modifying drugs (DMDs) may impair immune surveillance and increase susceptibility to infections, including TB [15,23].

MS is a chronic inflammatory autoimmune disease of the central nervous system. Therapy of MS involves DMDs with different mechanisms of action, such as monoclonal antibodies and small-molecule oral agents [24,25]. Immunomodulatory and immunosuppressive DMDs may increase patients’ susceptibility to various infections [26,27] by modulating the immune system [28,29,30,31]. Through the Delphi technique, a systematic process of forecasting, using the collective opinion of panel members [32], important recommendations have been listed to prevent infections in MS patients [23]. The Delphi consensus studies reported that the risk of TB disease in MS varies depending on the DMD [23,33]. The use of alemtuzumab, cladribine, and teriflunomide is associated with a slightly increased risk of developing TB as compared to the general population. However, this risk remains lower than that associated with TNF-α inhibitors, which are employed in the treatment of various immune-mediated inflammatory diseases other than MS [23,34,35,36,37,38,39,40]. No increased risk of TB disease was documented for interferons (IFNs), glatiramer acetate, dimethyl fumarate (DMF), fingolimod, natalizumab, and rituximab. Based on these findings, screening for tuberculosis infection (TBI) is recommended for patients initiating treatment with alemtuzumab, cladribine, and teriflunomide. Furthermore, such screening should be considered if these DMDs are potential future therapies [23,33,41].

No diagnostic gold standard exists for TBI diagnosis [42]. The tuberculin skin test (TST) and interferon-γ release assays (IGRAs) are the available commercial tests for detecting TBI [3]. They are both based on the evaluation of immune response after in vitro stimulation with Mtb antigens. Compared to TST, IGRAs have several advantages, such as higher specificity and the presence of internal positive and negative controls. However, since IGRAs are based on the detection of an immune response, they have reduced sensitivity in children and immune-compromised subjects such as people living with HIV or immune-mediated inflammatory disease (IMID) [3,43,44].

Immune-based tests for diagnosing TBI could give false negative results in MS patients under therapy [45] as already demonstrated in other fragile populations [44,46,47,48]. According to the Delphi consensus, TBI should be preferably investigated before starting biologic drugs or other immune-suppressive drugs to avoid the impact of this treatment on test scores [23].

It has been demonstrated that patients with MS under DMDs have a high risk of indeterminate results generated by QuantiFERON^®^-TB Gold In-Tube (QFT-GIT) [49,50]. Currently, the QFT-GIT has been replaced by the QuantiFERON^®^-TB Gold Plus (QFT-Plus) [3], including TB1 and T2 tubes eliciting the Mtb-specific immune response. The TB1 and TB2 tubes contain peptides derived from the Mtb antigens ESAT-6 and CFP-10 and can induce mainly CD4-reponse (TB1) or both CD4- and CD8-response (TB2) [3].

No data are available on the characterization of the immune response to the QFT-plus in MS populations. Therefore, in this prospective study, we aimed to compare the magnitude of the QFT-Plus response of patients with MS and TBI (MS-TBI) with TBI subjects without MS (NON-MS-TBI); additionally, discordant response to TB1/TB2 stimulation was reported. The results were evaluated considering the demographic and clinical data, particularly the impact of the specific DMDs and the type of TB exposure.

## 2. Materials and Methods

### 2.1. Study Participants

This prospective study was conducted at the National Institute for Infectious Diseases (INMI) L. Spallanzani and approved by the INMI Ethical Committee: approval no. 72/2015 (approved on 14 July 2015) and no. 27/2019 (approved on 15 July 2019). The study was conducted in collaboration with the Department of Neurosciences, San Camillo-Forlanini Hospital, in Rome.

Eight hundred ten patients with MS who regularly attended the MS Centre at S. Camillo-Forlanini Hospital were screened for TBI using the QFT-Plus assay between December 2018 and August 2023. All patient data were anonymized. TBI was diagnosed in 30 MS patients. Since only qualitative QFT-Plus results were initially available, the test was repeated for research purposes to obtain quantitative values. The assay was repeated in 20 MS-TBI patients who agreed to participate in the study, and written informed consent was obtained.

Subjects with TBI without MS (NON-MS-TBI) were also enrolled as a control group; their main features are described elsewhere [46]. Among them, 50 NON-MS-TBI subjects were defined as recently having TBI with documented exposure to Mtb in the previous 3 months; the remaining 54 TBI subjects had remote TBI since the time of exposure was unknown, and they performed the screening test for immigration reasons, job requirements, or adoption prerequisites. Patients with MS and a positive QFT-Plus score were sent from the neurologist (CG, SG, CT, SG) to the infectious disease specialist (DG) to have a chest radiograph and eventually exclude the presence of Mtb in the sputum. TBI diagnosis was based on a positive QFT-Plus score in the absence of radiological signs and clinical symptoms of TB disease. All MS-TBI patients were considered as remote TBI since the time of exposure was unknown.

### 2.2. QuantiFERON-TB-Plus (QFT-Plus)

QFT-Plus (Diasorin, Vercelli, Italy) was performed according to the manufacturer’s instructions. Plasma supernatants were collected and IFN-γ was measured by an ELISA. We defined the subjects responding only to one tube, TB1 or TB2, as “discordant responders”. The results were analyzed by QFT-Plus Analysis Software Version 2.71 (www.quantiFERON.com) and evaluated according to manufacturer’s criteria.

QFT-Plus results equal to or greater than 0.35 IU/mL were classified as positive. The selection of the QFT-Plus range (≥0.2 to ≤0.7 IU/mL) was based on multiple studies indicating that values within this interval are associated with relevant risk factors and/or evidence of TBI, as well as a higher likelihood of conversion upon repeated testing [48,51,52,53].

### 2.3. Statistical Analysis

Data analysis was performed using Prism 6 software (Graphpad Software 8.0, San Diego, CA, USA). Categorical measures were compared using a Chi-square test or Fisher’s exact test. For continuous measures, medians and interquartile ranges (IQRs) were reported and Kruskall–Wallis or Mann–Whitney U tests were used for comparisons across groups. Two-tailed *p*-values < 0.05 were considered significant.

## 3. Results

### 3.1. Demographic and Epidemiological Characteristics of the Population

Among the 810 patients with MS screened for TBI, 30 (3.7%, 95% confidence interval: 2.4–5.0) tested IGRA-positive; among them, 20/30 (67%) were recruited for the study. As a control group, we enrolled 106 TBI subjects without MS, defined as NON-MS-TBI. Women made up 54% of the population, and 53% of the enrolled subjects were from Western Europe, while 27% were from Eastern Europe; their median age was 42 years (Table 1). The demographic characteristics of the MS-TBI group did not differ from those of the NON-MS-TBI group, with the exception of a significant difference in the time of Mtb exposure, as 68% of the NON-MS-TBI reported a recent infection, whereas 100% of the MS-TBI patients were considered to have remote infection (*p* < 0.0001). Among the MS-TBI patients, seventeen accepted and completed the TB preventive therapy (TPT), one refused the TPT, and two subjects had already taken the TPT in the past. Among the MS-TBI group, 45% were under DMDs at the time of enrolment; most of them (93%) had relapsing–remitting MS; 45% had an Expanded Disability Status Scale between 3 and 4 and a lymphocyte count of 1.8 × 10^3^/mm^3^ (Table 2). None of the MS-TBI patients developed TB disease during the time of observation (at least 16 months after enrolment).

### 3.2. IFN-γ Response to QFT-Plus in TBI Subjects with or Without MS

A QFT-Plus assay was performed for each enrolled subject (Figure 1A). MS-TBI patients showed significantly lower IFN-γ production in response to TB1 and TB2 stimulation compared to NON-MS-TBI individuals (*p* = 0.01 and *p* = 0.02, respectively) (Figure 1A). All the participants responded to the mitogen, which served as the positive control.

Considering that MS-TBI subjects had remote exposure to Mtb whereas NON-MS-TBI patients included both recently and remotely exposed subjects [46], we compared the two groups, stratifying the patients according to the time of Mtb exposure (Figure 1B,C). Interestingly, while the MS-TBI subjects had a significantly lower amount of IFN-γ production compared to that from recent NON-MS-TBI in response to TB1 or TB2 (TB1: *p* = 0.05; TB2 *p* = 0.02), the difference was not significant if compared to remote NON-MS-TBI.

### 3.3. Detailed Analysis of the Response to TB1 and TB2 in TBI with or Without MS

Focusing on the 20 MS-TBI patients who scored positively for QFT-Plus, we found that 90% responded to TB1 and 90% to TB2 stimulation. Similarly, we found that 95% of TBI subjects responded to TB1 and 96% responded to TB2 stimulation (Table 3). Interestingly, MS-TBI patients had 30% of TB2 results falling in the grey zone of the QFT-Plus (0.2–0.7 IU/Ml) [46], whereas only 7% of NON-MS-TBI subjects showed this profile or response (*p* = 0.006) (Table 3). Stratifying the NON-MS-TBI according to the time of exposure to Mtb, we found that MS-TBI had a higher number of results falling in the grey zone compared to recent NON-MS-TBI (TB1: *p* = 0.04; TB2: *p* = 0.006) (Table 3). Differently, remote NON-MS-TBI showed a result distribution similar to MS-TBI patients (Table 3). When stratifying MS-TBI patients based on treatment exposure, we observed that DMDs did not influence the distribution of QFT-Plus results within the grey zone of the QFT-Plus (Table 4). A similar IFN-γ response was observed in MS-TBI patients receiving DMDs compared to those not receiving any therapy (Appendix A). There was no significant correlation between lymphocyte counts and IFN-γ response to TB1 and TB2 stimulation (Figure 2).

## 4. Discussion

This study, conducted in Italy, a country with low TB incidence, assessed the impact of MS on the QFT-Plus response in individuals with TBI. For comparative purposes, we included TBI individuals without MS as a control group.

Literature data on the prevalence of TBI in MS are limited to few observational studies with small sample sizes (ranging from 58 to 222 patients). The estimated 3.7% prevalence in our study is quite in line with a previous report from Germany (3.85%) [27], but superior to the 1.8% prevalence reported in United States [49], and significantly below the 29.3% reported in Brazil, a TB-endemic country [54]. In this regard, our study reinforces the recommendation to screen for TBI patients with MS before starting DMDs.

We found a significantly lower amount of IFN-γ released in response to TB1 and TB2 stimulation of the QFT-Plus in MS-TBI compared to NON-MS-TBI individuals. These results are supported by similar findings in immunocompromised subjects such as people living with HIV and patients with immune-mediated inflammatory disease taking immune suppressive drugs [44,46]. However, given the assumption that individuals with MS-TBI had previous exposure to Mtb, we observed that their QFT-Plus response was comparable to that of NON-MS-TBI individuals with remote Mtb exposure. These findings indicate that MS status itself does not influence the QFT-Plus response when results are categorized based on Mtb exposure.

However, MS status had a significant impact on the IFN-γ quantitative response to both TB1 and TB2 stimulation, with a high proportion of TB2 results falling within the grey zone of the QFT-Plus assay. These findings are in agreement with previous results in TBI subjects with IMID, such as rheumatoid arthritis [46]. By stratifying the NON-MS-TBI subjects according to the time of Mtb exposure, we demonstrated that remote exposure was associated with a higher probability of results falling in the grey zone. This suggests a sub-optimal evaluation of the Mtb-specific response in individuals previously exposed to Mtb. In contrast, when stratifying patients by the presence or absence of MS treatment, our findings indicated that the presence of MS therapy did not influence the distribution of IFN-γ values in MS-TBI patients.

Evidence suggests a link between grey zone results and the risk of developing TB disease in IMID patients [48]. Although the grey zone has been defined [48,51,52,53,55,56,57,58], its clinical interpretation in MS patients remains challenging. In the presence of borderline results, clinical decisions should rely on a multidisciplinary risk assessment—including input from an infectious diseases specialist—that considers individual TB risk factors and may include the repetition of the IGRA. Complementary diagnostic tools, including extended in vitro stimulation (such as long-term stimulation) or various immune-based biomarkers [59], are currently used primarily in research studies and have not been implemented in clinical practice. This approach aligns with current practices in other immunocompromised populations, where grey zone results have been associated with increased TB progression risk [48].

A correlation between the lymphocyte number and a positive QFT-Plus score has been shown [54]. In the present study, even though most MS-TBI patients were on immune-modulating therapy, the total lymphocyte count was in the normal number range and did not impact TB1 or TB2 response, and all subjects responded to mitogens.

This study focused on patients with MS who tested positive for the QFT-Plus assay and were subsequently referred to infectious disease specialists. The necessity for such referrals arises from the increased risk of TB progression associated with specific DMDs. We demonstrated a lower magnitude of IFN-γ response to Mtb antigens of the QFT-Plus in MS-TBI individuals. Our data indicate that QFT-Plus borderline results should be carefully considered to avoid false negatives due to MS status or exposure to DMDs. A long-term in vitro stimulation with Mtb antigens could recall the Mtb immune response [60] and induce a higher IFN-γ production, avoiding borderline results. New research tests based on immune RNA signatures [61,62], Mtb DNA detection [42], or immunological markers [12,63,64,65,66,67,68,69] could be instrumental, if validated, in identifying subjects truly infected [70].

This prospective study is limited by the relatively low number of enrolled patients and the diverse range of MS therapies considered. The administration schedules of DMDs vary significantly—ranging from weekly to monthly or even yearly dosing—making it difficult to evaluate the QFT-plus response according to the MS treatment duration across individuals. The majority of MS-TBI individuals (93%) had relapsing–remitting multiple sclerosis. Since only two patients had primary progressive MS, we did not stratify the results by MS subtype. However, both MS-TBI patients with primary progressive MS had QFT-Plus results outside the uncertainty range (>0.7 IU/mL) for both TB1 and TB2 stimulation. However, the study includes controls with varying exposure times, and the results provide important insights into the application of IGRAs in patients with MS.

In conclusion, we assessed the response to the QFT-Plus in Italy, a low-TB-endemic country, among patients with MS, diagnosed with TBI and eligible for DMDs.

Our findings show that individuals with MS-TBI have a significantly diminished IFN-γ response to the QFT-Plus when compared to NON-MS-TBI patients. Furthermore, there is an increased proportion of results within the borderline zone, irrespective of the type of MS treatment administered. These findings have relevant implications for clinical practice, particularly for clinicians who may be less familiar with the interpretation of TB testing in MS patients. The finding of a reduced IFN-γ response and more borderline QFT-Plus results in MS-TBI individuals indicates that the standard IGRA interpretation may underestimate TBI in those with negative QFT values within the “negative grey zone” (IFN-γ 0.2–0.34 IU). Therefore, screening protocols for MS patients—especially those who are candidates for immunosuppressive DMDs—should include careful evaluation of borderline results and early referral to infectious disease specialists. Enhanced monitoring may be warranted in cases with borderline results, even in low-TB-endemic settings, to prevent missed diagnoses and mitigate the risk of TB reactivation during immunosuppressive therapy.

Future studies will help characterize the real meaning of the borderline zone of the QFT-Plus, especially in vulnerable populations, to better define the accuracy of the assay in terms of sensitivity, specificity, and the prediction of TB development.

## Figures and Tables

**Figure 1 neurolint-17-00119-f001:**
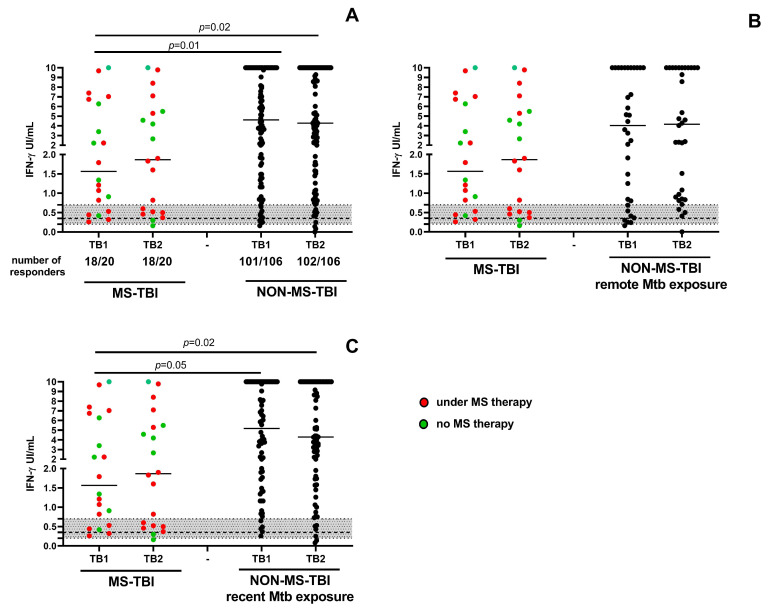
**Evaluation of IFN-γ production in response to QFT-Pus antigens in TBI subjects with or without MS.** IFN-γ production was evaluated by ELISA in response to QFT-Plus antigens. (**A**) MS-TBI vs. NON-MS-TBI individuals; (**B**) MS-TBI vs. remote NON-MS-TBI individuals; (**C**) MS-TBI vs. recent NON-MS-TBI individuals. The dotted line represents the cut-off value of 0.35 IU/mL. The grey section represents the IFN-γ values ranging in 0.2–0.7 IU/mL (“uncertainty zone”). Statistical analysis was performed using the Mann–Whitney U test, and the *p* value was considered significant if ≤0.05. Footnotes: TBI: tuberculosis infection; MS: multiple sclerosis; TB1: tube 1; TB2: tube 2; IU: international unit; IFN: interferon.

**Figure 2 neurolint-17-00119-f002:**
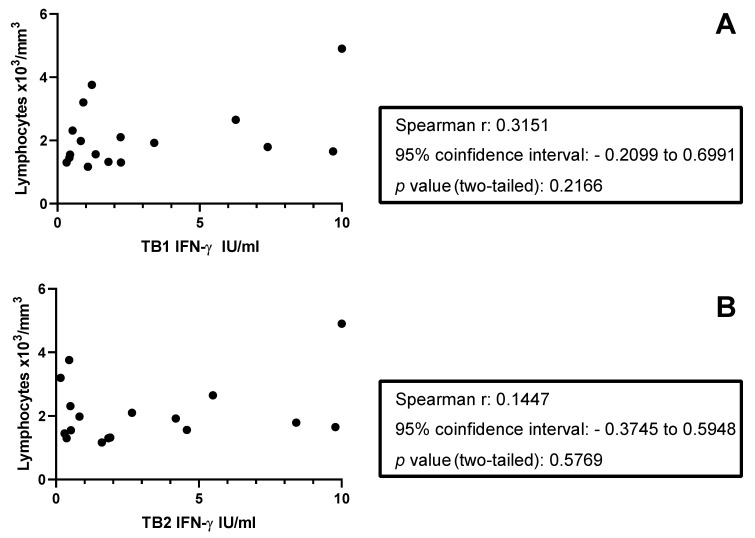
**Correlation of IFN-γ production in response to QFT-Plus stimulation and the number of lymphocytes in MS-TBI patients.** IFN-γ production was evaluated by ELISA in response to QFT-Plus antigens, the number of lymphocytes was available for 17/20 patients, and the lymphocyte count was performed not more than two months before enrolment. (**A**) Correlation between IFN-γ production in response to TB1 stimulation and number of lymphocytes; (**B**) correlation between IFN-γ production in response to TB2 stimulation and number of lymphocytes. Statistical analysis was performed using Spearman’s correlation test. Footnotes: TB1: tube 1; TB2: tube 2; IU: international unit; IFN: interferon.

**Table 1 neurolint-17-00119-t001:** Clinical and epidemiological characteristics of the TBI subjects enrolled.

	MS-TBI N = 20	NON-MS-TBI N = 106	Total N = 126	*p*
**Female, N (%)**	12 (60)	56 (50)	68 (54)	0.97 *
**Median age, IQR**	41 (35–57)	41 (35–56)	42 (28–55)	0.37 ^#^
**Origin, N (%)**				
**Est Europe, N (%)**	5 (25)	29 (27)	34 (27)	na **
**West Europe, N (%)**	12 (60)	55 (52)	67 (53)	
**South America, N (%)**	0 (0)	3 (3)	3 (2)	
**Asia, N (%)**	1 (5)	10 (9)	11 (9)	
**Africa, N (%)**	2 (10)	9 (9)	11 (9)	
**TB exposure, N (%)**				
**Recent**	0	72 (68)	72 (57)	*p* ≤ 0.0001 *
**Remote**	20 (100)	34 (32)	54 (43)	

**Footnotes:** MS: multiple sclerosis, TBI: tuberculosis infection, IQR: interquartile range; na: not applied; * Chi square test; ^#^ Mann–Whitney; ** test not applied because Chi-square calculations are only valid when all expected values are greater than 1.0 and at least 20% of the expected values are greater than 5. These conditions were not met.

**Table 2 neurolint-17-00119-t002:** Clinical and epidemiological characteristics of MS patients.

	MS-TBI N = 20
**Under MS therapy N (%)**	9 (45)
**Type of MS therapy**	
**Glatiramer acetate**	1 (11)
**IFN-** **β** **N (%)**	5 (56)
**Anti-CD20** **N (%)**	1 (11)
**Dimethyl fumarate** **N (%)**	1 (11)
**Teriflunomide** **N (%)**	1 (11)
**Primary progressive MS (%)**	2 (10)
**Relapsing–remitting MS (%)**	18 (93)
**Expanded Disability Status Scale**	
**0**	2 (10)
**1–2**	5 (25)
**3–4**	9 (45)
**5–6.5**	4 (20)
**Lymphocytes × 10^3^/mm^3^**	1.8 (1.5–2.3)

**Footnotes**: MS: multiple sclerosis; IFN: interferon.

**Table 3 neurolint-17-00119-t003:** QFT-Plus response among MS-TBI and NON-MS-TBI patients.

	QFT-Plus Tube	MS-TBI	NON-MS-TBI	*p **	NON-MS-TBI Recent Exposure	*p **	NON-MS-TBI Remote Exposure	*p **
Enrolled subjects N		20	106		72		34	
**Number of responders** **N (%)**	**TB1**	18 (90)	101 (95)	0.30	71 (99)	0.12	30 (88)	>0.99
	**TB2**	18 (90)	102 (96)	0.24	69 (96)	0.30	33 (97)	0.55
**Number of results falling in the grey zone 0.2–0.7 IU/mL** **N (%)**	**TB1**	5 (25)	12 (11)	0.15	5 (7)	** *0.04* **	7 (21)	0.7
	**TB2**	6 (30)	7 (7)	** *0.006* **	4 (6)	** *0.006* **	3 (9)	0.06

**Footnotes:** MS: multiple sclerosis, TBI: tuberculosis infection, * Fisher test of NON-MS-TBI (total, recent, or remote) against MS-TBI patients.

**Table 4 neurolint-17-00119-t004:** QFT-Plus results falling in the grey zone among the MS-TBI patients stratified according to MS therapy.

	QFT-Plus Tube	MS-TBI Under MS Therapy N = 9	MS-TBI NO-MS Therapy N = 11	*p* *
**Number of results falling in the grey zone 0.2–0.7 IU/mL N (%)**	**TB1**	3 (33)	2 (18)	0.6169
	**TB2**	3 (33)	3 (27)	>0.9999

**Footnotes:** MS: multiple sclerosis, TBI: tuberculosis infection, * Fisher test.

## Data Availability

The raw data are available in our institutional repository (rawdata.inmi.it), subject to registration. The data can be found by selecting the article of interest from a list of articles ordered by year of publication. No charge for granting access to data is required. In the event of a malfunction of the application, the request can be sent directly by e-mail to the Library (biblioteca@inmi.it).

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
