# Peer review of "Characterization of QuantiFERON-TB-Plus Results in Patients with Tuberculosis Infection and Multiple Sclerosis"

_2035-8377, 2025, doi:10.3390/neurolint17080119_

Round 1
Reviewer 1 Report
Comments and Suggestions for Authors
This is a well-structured, methodologically sound, and timely manuscript that investigates the characterization of QuantiFERON-TB-Plus (QFT-Plus) responses in patients with multiple sclerosis (MS) and latent tuberculosis infection (TBI). It addresses an important gap in the literature regarding TB screening in immunocompromised populations, particularly MS patients undergoing disease-modifying therapies (DMDs). The results are presented clearly and supported by appropriate statistical analysis. The study also provides practical implications for interpreting grey-zone results in vulnerable populations.
I only have minor comments for the manuscript:
1. Clarify terminology consistency: Use uniform terminology throughout (e.g., "QFT-Plus" vs. "QFT-Pus" appears once in Figure 1 legend and should be corrected).
2. Borderline/grey zone: Although the grey zone is defined and referenced, the discussion could briefly elaborate on how clinical decisions should be guided in the presence of such borderline results in MS patients.
3. Expand on clinical implications: The conclusions would benefit from a short mention of how these findings may influence screening recommendations or patient monitoring, especially for clinicians unfamiliar with the nuances of TB testing in MS.
4. Formatting: Ensure that all footnotes in tables are placed consistently and clearly (e.g., Table 3).
5. Reference update: The reference list is strong and up to date. However, ensure that all citations conform to MDPI style and that the formatting of DOIs is consistent (some are written with "doi:", others embedded as URLs)
Author Response
Referee 1:
This is a well-structured, methodologically sound, and timely manuscript that investigates the characterization of QuantiFERON-TB-Plus (QFT-Plus) responses in patients with multiple sclerosis (MS) and latent tuberculosis infection (TBI). It addresses an important gap in the literature regarding TB screening in immunocompromised populations, particularly MS patients undergoing disease-modifying therapies (DMDs). The results are presented clearly and supported by appropriate statistical analysis. The study also provides practical implications for interpreting grey-zone results in vulnerable populations.
I only have minor comments for the manuscript:
- Clarify terminology consistency: Use uniform terminology throughout (e.g., "QFT-Plus" vs. "QFT-Pus" appears once in Figure 1 legend and should be corrected).
Thank you for the comment. We have corrected the terminology to ensure consistency throughout the manuscript, including the legend of Figure 1
- Borderline/grey zone: Although the grey zone is defined and referenced, the discussion could briefly elaborate on how clinical decisions should be guided in the presence of such borderline results in MS patients.
Thank you for the comment. We have updated the discussion section to further elaborate on how clinical decisions should be guided in the presence of borderline QFT-Plus results in MS patients.
- Expand on clinical implications: The conclusions would benefit from a short mention of how these findings may influence screening recommendations or patient monitoring, especially for clinicians unfamiliar with the nuances of TB testing in MS.
Thank you for the comment. We have updated the conclusions of the manuscript to include a brief discussion on how these findings may influence screening recommendations and patient monitoring, particularly for clinicians less familiar with TB testing in MS patients
- Formatting: Ensure that all footnotes in tables are placed consistently and clearly (e.g., Table 3).
Thank you for the comment. We have corrected the footnotes to ensure consistent and clear formatting across all tables, including Table 3
- Reference update: The reference list is strong and up to date. However, ensure that all citations conform to MDPI style and that the formatting of DOIs is consistent (some are written with "doi:", others embedded as URLs)
Thank you for the comment. We have updated the reference list according to the Neurology International style guidelines using Zotero, ensuring consistency in DOI formatting throughout.

Reviewer 2 Report
Comments and Suggestions for Authors
The study by Petruccioli Characterization of QuantiFERON-TB-Plus results in patients with tuberculosis infection and multiple sclerosis addresses an important clinical issue — assessing the effectiveness of the QuantiFERON-TB-Plus test in patients with MS and TBI. The research is relevant due to the need to optimize tuberculosis diagnostics in immunocompromised patients.
Recommendations: What stages of multiple sclerosis were the patients in during the study? This information is crucial for understanding how disease severity might influence test results.
The study shows that MS patients had results falling in the “grey zone” of the test. Could the authors provide a more detailed analysis of the criteria for the grey zone in these patients?
The article is recommended for publication after revision, taking into account these comments.
Author Response
The study by Petruccioli Characterization of QuantiFERON-TB-Plus results in patients with tuberculosis infection and multiple sclerosis addresses an important clinical issue — assessing the effectiveness of the QuantiFERON-TB-Plus test in patients with MS and TBI. The research is relevant due to the need to optimize tuberculosis diagnostics in immunocompromised patients.
- Recommendations: What stages of multiple sclerosis were the patients in during the study? This information is crucial for understanding how disease severity might influence test results.
Thank you for the comment. This information is included in Table 2. However, we have also updated Section 3.1 of the Results to clarify the clinical status of MS patients.
- The study shows that MS patients had results falling in the “grey zone” of the test. Could the authors provide a more detailed analysis of the criteria for the grey zone in these patients.
Thank you for the comment. We have updated the Methods section to provide a more detailed explanation of the criteria used to define the grey zone in MS patients.
The article is recommended for publication after revision, taking into account these comments.

Reviewer 3 Report
Comments and Suggestions for Authors
The authors studied the results of the QuantiFERON-TB-Plus test in patients with tuberculosis infection and multiple sclerosis. The work evaluates the effectiveness of this test. The advantage of this study is that it is prospective and novel.
There are some minor comments:
1. It is necessary to add a little about the comorbidity of tuberculosis and autoimmune diseases.
2. Why did the authors not take into account the type of multiple sclerosis in the results of QFT-PLUS? It should be added.
3. The study had many results in the gray zone (30%). Were alternative methods for diagnosing tuberculosis carried out? Or a repeat test?
4. The non-ms-tbi group has heterogeneity - it is worth mentioning this in the limitations. In the control group, there is no data on comorbid diseases and medications taken, for example, they could have taken glucocorticoids. In addition, there are many recently infected people in the control.
5. The authors state that there is no effect of therapy, but the duration of therapy is not specified and more than half of the patients were taking interferons, which have little effect on the immune response. This should be added to the discussion.
6. The authors should more precisely indicate the clinical significance and specific clinical recommendations.
Author Response
The authors studied the results of the QuantiFERON-TB-Plus test in patients with tuberculosis infection and multiple sclerosis. The work evaluates the effectiveness of this test. The advantage of this study is that it is prospective and novel.
There are some minor comments:
1. It is necessary to add a little about the comorbidity of tuberculosis and autoimmune diseases.
Thanks for the comment, we updated the introduction as suggested.
- Why did the authors not take into account the type of multiple sclerosis in the results of QFT-PLUS? It should be added.
Thank you for the comment. The majority of MS-TBI individuals (93%) had relapsing-remitting multiple sclerosis. Since only two patients had primary progressive MS, we decided not to stratify the results according to MS type. However, for the reviewer’s information, both MS-TBI patients with primary progressive MS had QFT-Plus results >0.7 IU/mL for both TB1 and TB2 stimulation. We added it in the text
- The study had many results in the gray zone (30%). Were alternative methods for diagnosing tuberculosis carried out? Or a repeat test?
Thank you for the comment. Although the grey zone is defined and referenced, its clinical interpretation in MS patients remains challenging.
For borderline results, clinical decisions should rely on a multidisciplinary risk assessment—including input from an infectious diseases specialist—that considers individual TB risk factors and may include repeating the IGRA.
Complementary diagnostic tools, including extended in vitro stimulation (such as long-term stimulation) or various immune-based biomarkers, are currently used primarily in research studies and have not been implemented in clinical practice.
This explanation has been added to the discussion of the manuscript.
- The non-ms-tbi group has heterogeneity - it is worth mentioning this in the limitations. In the control group, there is no data on comorbid diseases and medications taken, for example, they could have taken glucocorticoids. In addition, there are many recently infected people in the control.
Thank you for your comment. Individuals with autoimmune diseases other than MS were excluded from the study. Information regarding therapies used by the control group during the screening period was not collected. If TBI subjects received corticosteroid therapy, it would likely have been temporary and at a low dosage, as autoimmune diseases were exclusion criteria for study enrolment.
We acknowledge that most enrolled TBI subjects have had recent contact with individuals diagnosed with TB disease. This results from current screening practices, which primarily target this patient population in accordance with WHO guidelines. Cases of remote TB infection are identified predominantly within specific groups, such as candidates for biological therapy, HIV-infected individuals, and prisoners.
Therefore, we consider the comparison valid, as this distribution accurately reflects the epidemiology of TBI in Italy—a country with low endemicity and limited circulation of Mycobacterium tuberculosis.
- The authors state that there is no effect of therapy, but the duration of therapy is not specified and more than half of the patients were taking interferons, which have little effect on the immune response. This should be added to the discussion.
We acknowledge the reviewer’s comment regarding the lack of information on therapy duration. However, MS patients are typically on long-term, often lifelong, treatment regimens. The administration schedules of DMDs vary significantly—ranging from weekly to monthly or even yearly dosing—making it difficult to evaluate the QFT-plus response according to the MS treatment duration across individuals. Moreover, the MS group is limited. For these reasons, we chose to describe the type of MS therapy rather than stratify the results based on MS treatment duration. We added this explanation in the discussion.
- The authors should more precisely indicate the clinical significance and specific clinical recommendations.
Thank you for the comment, we updated the discussion as requested.

Reviewer 4 Report
Comments and Suggestions for Authors
Dear Editor and Authors,
The manuscript entitled "Characterization of QuantiFERON-TB-Plus results in patients with tuberculosis infection and multiple sclerosis" is an original paper regarding the characterization of QFT-Plus response in patients with multiple sclerosis. The paper is well written, the methods and results are well presented, the references are appropriate. It would be of interest to readers.
I have only one remark: figure 1 should be enlarged to be better visible to readers.
Number of patients with MS and TB is quite small, but it has been stated by authors. I think the manuscript could be published after improvement of fig, 1.
Best greetings,
Author Response
The manuscript entitled "Characterization of QuantiFERON-TB-Plus results in patients with tuberculosis infection and multiple sclerosis" is an original paper regarding the characterization of QFT-Plus response in patients with multiple sclerosis. The paper is well written, the methods and results are well presented, the references are appropriate. It would be of interest to readers.
I have only one remark: figure 1 should be enlarged to be better visible to readers.
Thank you for the comment. In the final version of the article, we will upload the figure in TIFF format, which should enhance its clarity and resolution. In the meantime, we have modified the current version of the figure to improve its visibility for readers.
Number of patients with MS and TB is quite small, but it has been stated by authors. I think the manuscript could be published after improvement of fig, 1.
